# Self-suppressing behavioral patterns and depressive traits exacerbate chronic pain: Psychological trait assessment using the structured association technique method

Shin Hashizume[1], Masako Nakano[1]*, Chihiro Ikehata[1], Nobuaki Himuro[2], Kanna Nagaishi[1,3], Mineko Fujimiya[1,3]

1 Department of Anatomy, Sapporo Medical University School of Medicine, Sapporo, Hokkaido, Japan, 2 Department of Public Health, Sapporo Medical University School of Medicine, Sapporo, Hokkaido, Japan, 3 Rene Clinic Tokyo, Chiyoda, Tokyo, Japan

* m.nakano@sapmed.ac.jp

**Data availability statement:** All relevant data are within the manuscript and its Supporting Information files.

**Funding:** The author(s) received no specific funding for this work.

**Competing interests:** The authors have declared that no competing interests exist.

## Abstract

This study investigated the relationship between psychological traits and chronic pain using the Structured Association Technique (SAT) method to evaluate psychological factors associated with chronic pain. The participants included 105 older adults (23 men, 82 women, mean age 80.82 years) who received rehabilitation services. Chronic pain severity was assessed using a numerical rating scale (NRS), and psychological traits were evaluated by SAT. In addition, maternal attachment experiences in childhood were examined. The NRS showed significant positive correlations with the self-suppressing behavioral pattern (S) scale ($r = 0.31$, $p = 0.001$), and the depression (D) scale ($r = 0.31$, $p = 0.001$). The proportion of participants with high scores on both the S and D scales (SD group) was notably higher in the high NRS group. Logistic regression analysis showed that the SD group had a higher odds ratio (OR = 8.469, $p = 0.004$) for severe chronic pain, suggesting that SD traits independently contribute to worse pain. In the SD group, the self-denial scale scores were high, and self-denial traits showed a negative correlation with maternal attachment experiences in childhood. This finding indicates that poor maternal attachment may enhance self-denial traits, which in turn indirectly worsen pain through their effects on S and D traits. The results of this study highlight the importance of S and D traits as psychological factors in chronic pain, particularly in Japanese populations, and suggest that assessing self-suppressing behavioral patterns may be beneficial for pain management. However, the cross-cultural validity of the SAT scales requires further investigation. SAT therapy may provide a comprehensive approach to the treatment and prevention of complex conditions influenced by psychological and social factors, including chronic pain.

## Introduction

Chronic pain is defined as pain that persists for longer than 3 months and causes a significant health problem. Chronic pain affects 20-50% of the global population, with a prevalence of approximately 40% in Japan [1]. Chronic pain has a negative effect on activities of daily living

and worsens quality of life [2]. In addition, the economic cost of chronic pain is substantial. The annual healthcare cost for chronic pain is estimated to be between $560 billion and $635 billion in the United States [3].

Chronic pain is a complex condition influenced by physical, psychological and social factors [4]. Of the psychological factors, depression is known to be the most common comorbidity associated with chronic pain [5]. In addition, anxiety disorders, such as panic disorder and post-traumatic stress disorder, also show a positive relationship with chronic pain [6]. Pharmacological therapies, such as acetaminophen and nonsteroidal anti-inflammatory drugs (NSAIDs), are typically used to manage the physical aspect of chronic pain. However, these treatments often fail to achieve complete recovery because they do not address the psychological factors involved. Recently, cognitive functional therapy (CFT) for chronic pain has been developed [7,8]. CFT is a type of cognitive behavioral therapy (CBT) that targets three components, cognitive component, functional component, and lifestyle component [9]. Patients with chronic pain often have a negative cognitive component, as an excessive fear to move the pain area. They also have an abnormal functional component, such as hypertonic muscular contraction. In addition, they often have an unhealthy lifestyle, such as insufficient exercise and lack of sleep. CFT targets these abnormal components by correcting the wrong cognition, abnormal movements, and unhealthy habits. However, effective CBTs targeting the complex factors associated with chronic pain are yet to be well established.

The Structured Association Technique (SAT), a type of CBT, was developed in the 1990s by T. Munakata [10]. SAT therapy uses structured questions to elicit imaginative associations and insights in patients. These associations help patients recognize underlying issues in their minds and foster resilience by altering their interpretations of these issues. Munakata proposed a theory suggesting that congenital factors, the subconscious mind, and emotional development during childhood influence one's way of living in adulthood. The SAT method assesses patients' psychological and genetic characteristics using a designated questionnaire that evaluates thirteen psychological scales and six genetic characteristics [10]. The questionnaire helps to uncover a gap between their subconscious and current mental states. SAT therapy aims to correct these gaps and induces the patients to align with their innate sense of self. Other CBTs usually focus on correcting the inappropriate way of thinking, but SAT therapy rather targets the way of feeling. The past negative feeling is influenced by congenital and genetic factors, as well as childhood experience. However, the SAT method can change the past image memory into a positive image by using the image-script, which consists of sensory and emotional information [11].

Previously, SAT therapy was used for children with psychogenic visual disturbance (PVD) [12]. Results from the SAT questionnaire showed that the PVD group had higher scores on the self-suppressing behavioral pattern scale and interpersonal dependency scale, and lower scores on the self-worth scale, indicating high stress levels in these patients. After undergoing SAT therapy, not only did their PVD symptoms improve, but their scores on these psychological scales also showed improvement. Because there is no established treatment for PVD, SAT therapy may offer an advantage over other CBTs [13]. However, there are no research reports of the application of SAT therapy for patients with chronic pain. In this study, the aim was to evaluate whether psychological trait assessments using the SAT method could serve as a useful tool for assessing the risk of chronic pain exacerbation.

## Methods

### Participants

Participants were recruited between August 1 2021 and October 1 2023 from two rehabilitation service centers, the Rihaport minami in Sapporo City (Facility 1) and the day service

center Karin in Hakodate City (Facility 2), Hokkaido, Japan. After being informed about the study, participants provided written, informed consent. People who meet the following inclusion criteria participated in this study: (1) age > 20 years old, and (2) no pain or non-cancer related chronic pain that persists for more than 3 months. People who meet the following exclusion criteria did not participate in this study: (1) acute pain that persists for less than 3 months, and (2) cognitive impairment. Finally, a total of 105 older adults (male 23, female 82) with an average age of 80.82 years participated in this study.

### Procedure and ethical approval

This study was approved by the Ethics Committee of Sapporo Medical University (Approval No. 3-1-19). All participants signed informed consent forms before the study. This study involving human participants was in accordance with the 1964 Helsinki Declaration and its later amendments or comparable ethical standards.

### Numerical rating scale

The severity of chronic pain lasting more than three months was assessed using the NRS. The NRS is an eleven-point scale ranging from zero (no pain) to ten (maximum pain experienced), which allows participants to indicate the intensity of their current pain. Participants were also asked about the medical history, the specific locations of their pain, the number of the pain locations, and the pain duration. For those reporting pain in multiple areas, the NRS score for each site was evaluated, and the highest NRS score was used to analyze the association between pain and psychological factors.

The NRS score is generally categorized into four groups: NRS 0 = no pain, NRS 1-3 = mild pain, NRS 4-6 = moderate pain, and NRS 7-10 = severe pain [14]. To investigate the psychological traits associated with severe pain, participants were divided into two groups: high NRS group (NRS: 7–10) and low NRS group (NRS: 0–6).

### Structured association technique method

The SAT scales were also evaluated. Of the 13 SAT scales, seven scales were assessed in this study: self-worth scale, self-suppressing behavioral pattern scale, interpersonal dependency scale, anxiety tendency scale, depression scale, self-denial scale, and post-traumatic stress syndrome (PTSS) scale. The remaining six scales (emotional support network, problem-solving behavioral pattern scale, SAT therapy requirement scale, emotional cognitive difficulty scale, self-pity scale, and self-dissociation scale) were not evaluated because the preliminary analysis suggested no association with NRS (S1 Fig). Each SAT scale was categorized into four levels (low, moderate, high, and very high) based on the scores (S1 Table). In addition, the degree of maternal attachment in childhood, which is believed to influence psychological traits assessed by the SAT method, was also investigated. Maternal attachment levels were classified into three categories, enough attachment (Score 0), a little attachment (Score 1), and poor attachment (Score 2). Participants were asked about the extent of love received from their mother in childhood, and they chose Score 0, Score 1, or Score 2.

### Statistical analysis

Statistical analyses were conducted using Spearman's correlation coefficient to evaluate the associations between the NRS score and the SAT psychological scales (R version 3.6.1; The R Foundation for Statistical Computing, Vienna, Austria). The distribution of the diagnosed and not diagnosed groups in the high and low NRS groups were analyzed using a Chi-squared test (R version 3.6.1). Data on psychological scales in the high and low NRS groups were analyzed

using an unpaired $t$-test (R version 3.6.1). The psychological group distributions in the high and low NRS groups were analyzed using a Chi-squared test (R version 3.6.1). The psychological groups are following: (1) both high self-suppressing behavioral pattern and high depression scales (High S, High D: SD group); (2) high self-suppressing behavioral pattern scale only (High S, Low D: S group); (3) high depression scale only (Low S, High D: D group); and (4) neither scale high (Low S, Low D: no-SD group). A logistic regression model (SPSS Statistics 25) was used to calculate odds ratios for the severity of chronic pain based on age, sex, facility, and the presence of SD, S, or D. Data on psychological scales across the SD, S, D, and no-SD group were analyzed using one-way ANOVA followed by Tukey's post hoc comparison test (R version 3.6.1). Spearman's correlation coefficient was also used to evaluate the association between each psychological scale and the degree of maternal attachment. Significance was set at $p < 0.05$, and data are presented as mean ± standard deviation (SD) values. Figures were created using GraphPad Prism 6.0 (GraphPad Software Inc., San Diego, CA, USA).

## Results

### Attributes and characteristics of the study participants

The participants' age, sex, body mass index (BMI), weekly exercise time, and numerical rating scale (NRS) scores, duration of pain, and the numbers of taking smoking and alcohol are shown in Table 1. The attributes of each facility (facility 1 and facility 2) are presented in S2 Table. The age of the participants was significantly lower at facility 1 than at facility 2, and the weekly exercise time for participants was significantly longer at facility 1 than at facility 2 (S2 Table).

### Pain areas

The pain areas reported by participants are shown in Table 2. The most common pain site was the lower back, followed by the knee and shoulder (Table 2). In addition, the number of chronic pain areas per participant is shown in S3 Table. Fifty-eight percent of participants reported chronic pain in two or more areas. Therefore, the pain areas listed in Table 2 included overlaps. The number of participants diagnosed or not diagnosed in the low and high NRS group are also shown in Table 2. No significant differences were observed between the number of diagnosed and not diagnosed groups in each pain area (Table 2).

**Table 1. Demographics of the participants.**

| (Mean ± SD) | All participants (n = 105) | Without chronic pain (n = 15) | Chronic pain (n = 90) |
|---|---|---|---|
| **Age** | 80.82 ± 8.57 | 82.8 ± 9.1 | 80.49 ± 8.48 |
| **Gender (men/ women)** | n = 23 (21.9%)/ n = 82 (78.1%) | n = 7 (46.7%)/ n = 8 (53.3%) | n = 16 (17.8%)/ n = 74 (82.2%) |
| **BMI** | 22.58 ± 4.45 | 23.76 ± 6.66 | 22.38 ± 3.99 |
| **Exercise intensity (min per week)** | 90.19 ± 153.62 | 51.0 ± 60.72 | 96.72 ± 163.38 |
| **NRS** | 4.64 ± 2.65 | 0 | 5.41 ± 1.99 |
| **Duration of pain** 3–6 months 6–12 months More than 12 months | – – – | 0 0 0 | n = 7 (7.8%) n = 10 (11.1%) n = 73 (81.1%) |
| **Smoking** | n = 4 (3.8%) | n = 1 (6.7%) | n = 3 (3.3%) |
| **Alcohol** | n = 21 (20.0%) | n = 2 (13.3%) | n = 19 (21.1%) |

The data for age, sex, BMI, exercise time per week, NRS score, duration of pain, smoking and alcohol for the study participants are shown. Values are the means ± SD.

**Table 2. Pain areas of the study participants.**

| Pain area | Total number | Low NRS (NRS 1-6) | | High NRS (NRS 7-10) | | p value |
|---|---|---|---|---|---|---|
| | | Diagnosed | Not Diagnosed | Diagnosed | Not Diagnosed | |
| **Lower-back** | n = 47 (44.8%) | n = 12<br><br>Lumbar spondylosis (n = 8)<br>Spinal canal stenosis (n = 4) | n = 20 | n = 6<br><br>Lumbar spondylosis (n = 4)<br>Spinal canal stenosis (n = 1)<br>Lumbar Disc Herniation (n = 1) | n = 9 | p = 0.87 |
| **Knee** | n = 35 (33.3%) | n = 10<br><br>Osteoarthritis (n = 10) | n = 13 | n = 3<br><br>Osteoarthritis (n = 3) | n = 9 | p = 0.28 |
| **Shoulder** | n = 23 (21.9%) | n = 1<br><br>Rotator cuff tear (n = 1) | n = 17 | n = 0 | n = 5 | p = 0.59 |
| **Other** | n = 23 (21.9%) | n = 8<br><br>Temporomandibular disorder (n = 1)<br>Carpal canal syndrome (n = 1)<br>Wrist fracture (n = 1)<br>Tenosynovitis (n = 1)<br>Rheumatoid arthritis (n = 2)<br>Achilles tendon rupture (n = 2) | n = 12 | n = 1<br><br>Cervicobrachial syndrome (n = 1) | n = 2 | p = 0.83 |

The number of participants who have pain in lower-back, knee, shoulder, and other areas are shown. The number of participants who were diagnosed or not diagnosed in the low and high NRS groups are also shown. The names of diseases are listed in the columns below of the diagnosed group. The distribution of the diagnosed and not diagnosed groups in the high and low NRS groups were analyzed using a chi-squared test.

## Correlations between the NRS and SAT psychological trait scales

The correlations between the NRS and the SAT psychological trait scales were evaluated. For this analysis, the NRS score at which participants reported the most intense pain, regardless of the pain site, was used. Although the SAT method evaluates psychological traits across 13 scales, this study focused on seven scales (self-suppressing behavioral pattern scale, depression scale, anxiety tendency scale, self-denial scale, PTSS scale, interpersonal dependency scale, and self-worth scale), based on preliminary findings suggesting their relevance to chronic pain. The analysis showed significant positive correlations between the NRS score and the self-suppressing behavioral pattern scale ($r = 0.31$, $p = 0.001$) and the depression scale ($r = 0.31$, $p = 0.001$) (Fig 1).

No significant correlations were observed for the other six scales, including the emotional support network scale (familial or not familial), problem-solving behavioral pattern scale, SAT therapy requirement scale, emotional cognitive difficulty scale, self-pity scale, and self-dissociation scale, as indicated in the preliminary study (S1 Fig). Similarly, no correlation was found between the NRS score and six genetic temperament traits (circulatory temperament, stickiness temperament, autistic temperament, obsessive temperament, anxiety temperament, and novelty-seeking temperament) (S2 Fig).

## Comparison of the SAT scales between the high and low NRS groups

The results showed that the scores of the self-suppressing behavioral pattern scale and the depression scale were significantly higher in the high NRS group than in the low NRS group ($p < 0.01$ and $p < 0.05$, respectively; Fig 2). No significant differences were observed for the 4 other scales, and no correlation was found between the self-suppressing behavioral pattern scale and the depression scale (Fig 3).

## Correlations of the self-suppressing behavioral pattern and depression scales with other SAT scales

To evaluate which scales affect the self-suppressing behavioral pattern scale and the depression scale, the correlations of these scales with the other five scales were analyzed. The

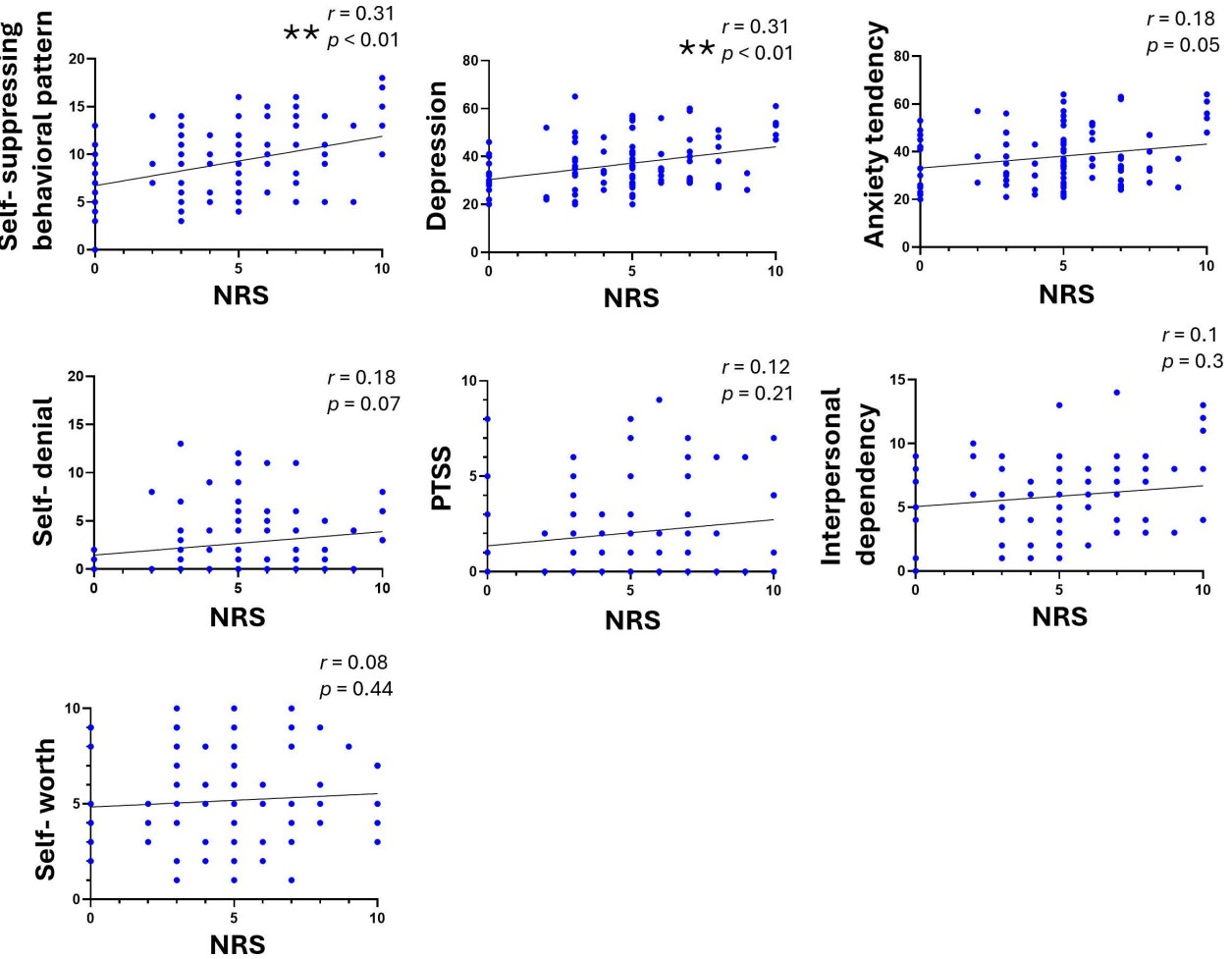

**Fig 1. The correlations between the NRS score and each scale in SAT therapy.** The correlations between the NRS score and each scale in SAT therapy (the self-suppressing behavioral pattern scale, the depression scale, the anxiety tendency scale, the self-denial scale, the PTSS scale, the interpersonal dependency scale, and the self-worth scale) are shown (n = 105). Spearman's correlation coefficient was used. **$p < 0.01$.

self-suppressing behavioral pattern scale showed significant correlations with the self-denial scale ($r = 0.34$, $p < 0.01$), the interpersonal dependency scale ($r = 0.30$, $p < 0.01$), and the anxiety tendency scale ($r = 0.24$, $p < 0.05$) (S3 Fig). Similarly, the depression scale showed significant correlations with the self-denial scale ($r = 0.59$, $p < 0.01$), anxiety tendency scale ($r = 0.51$, $p < 0.01$), PTSS scale ($r = 0.38$, $p < 0.01$), and self-worth scale ($r = 0.23$, $p = 0.02$) (S3 Fig).

## Combined influence of the self-suppressing behavioral pattern and depression traits on NRS scores

Based on the severity classification of each scale (S1 Table), participants were categorized into four groups: (1) both high self-suppressing behavioral pattern and high depression scales (High S, High D: SD group); (2) high self-suppressing behavioral pattern scale only (High S, Low D: S group); (3) high depression scale only (Low S, High D: D group); and (4) neither scale high (Low S, Low D: no-SD group). The proportions of these groups in the high and low NRS groups were compared.

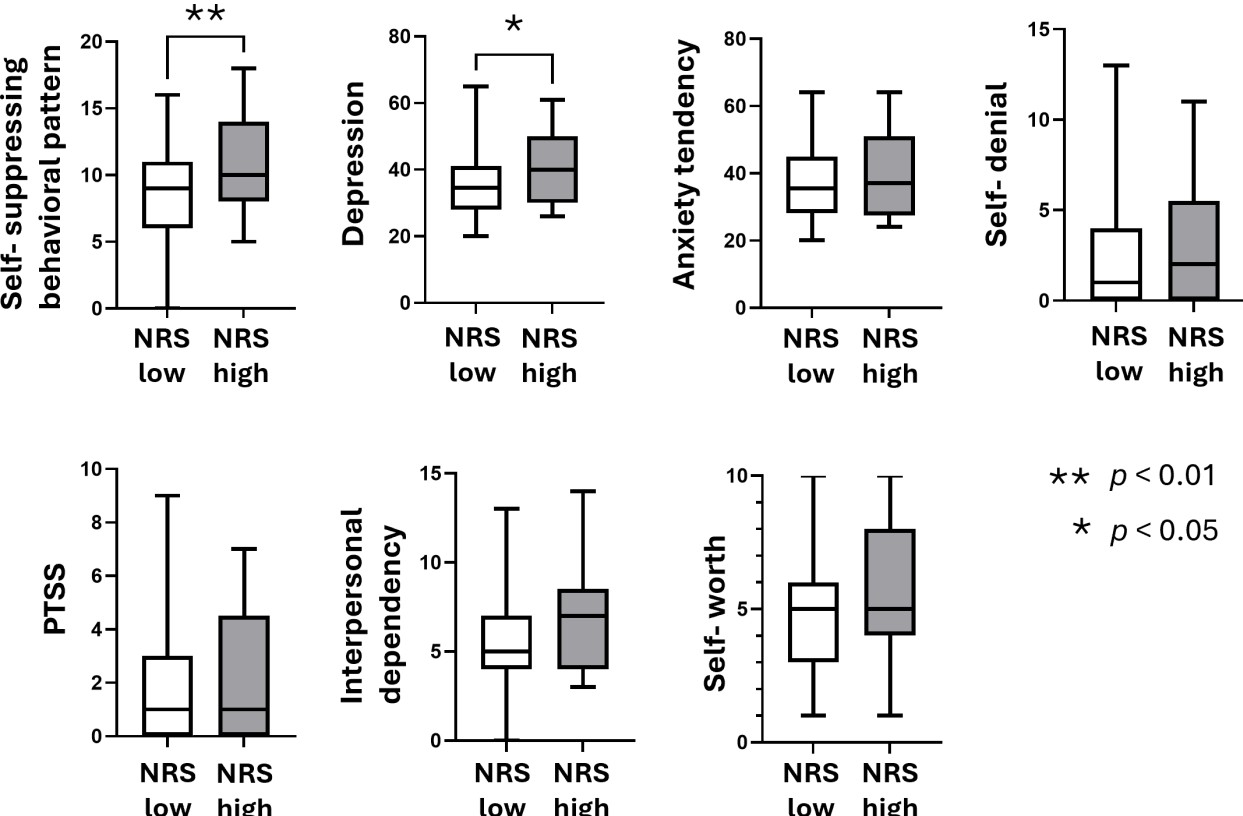

**Fig 2. Comparison of each scale in SAT therapy between the high and the low NRS score groups.** The comparisons of each scale in SAT therapy (the self-suppressing behavioral pattern scale, the depression scale, the anxiety tendency scale, the self-denial scale, the PTSS scale, the interpersonal dependency scale, and the self-worth scale) between the high NRS score group (NRS: 7-10, n = 25) and the low NRS score group (NRS: 0-6, n = 80) are shown. The data are shown as boxplots with legs (min and max values), and the unpaired *t*-test was used. * *p* < 0.05, ** *p* < 0.01.

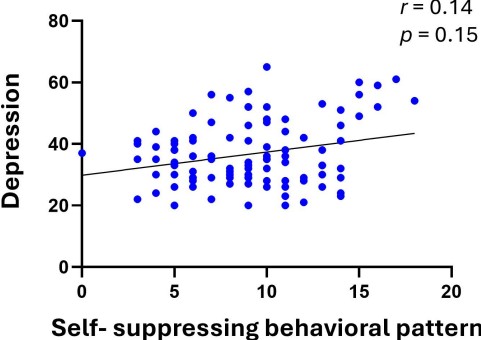

**Fig 3. Correlations between the self-suppressing behavioral pattern scale and the depression scale.** The self-suppressing behavioral pattern scale showed no significant correlation with the depression scale. Spearman's correlation coefficient was used (n = 105).

The SD group accounted for 11.2% in the low NRS group and 36% in the high NRS group. The S group accounted for 17.5% in the low NRS group and 12% in the high NRS group. The D group accounted for 32.5% in the low NRS group and 24% in the high NRS group. The

no-SD group accounted for 38.8% in the low NRS group and 28% in the high NRS group. Significant differences were observed between these groups (chi-squared test, $p = 0.04$; Fig 4).

## Logistic regression analysis for the factors contributing to NRS severity

The results showed a significant odds ratio of 8.469 for the SD group ($p = 0.004$, 95% CI: 2.011–35.664). No significant differences were found for the S group (OR: 1.457, $p = 0.646$, 95% CI: 0.293–7.249) or the D group (OR: 1.128, $p = 0.849$, 95% CI: 0.326–3.906) (Table 3). These findings suggest that the coexistence of high scores for self-suppressing behavioral patterns and depression is a risk factor for severe chronic pain.

## Relationship of maternal attachment in childhood with NRS and SAT scales

To investigate whether maternal attachment in childhood affects chronic pain, the correlation between the maternal attachment scores and the NRS score was analyzed, but no significant

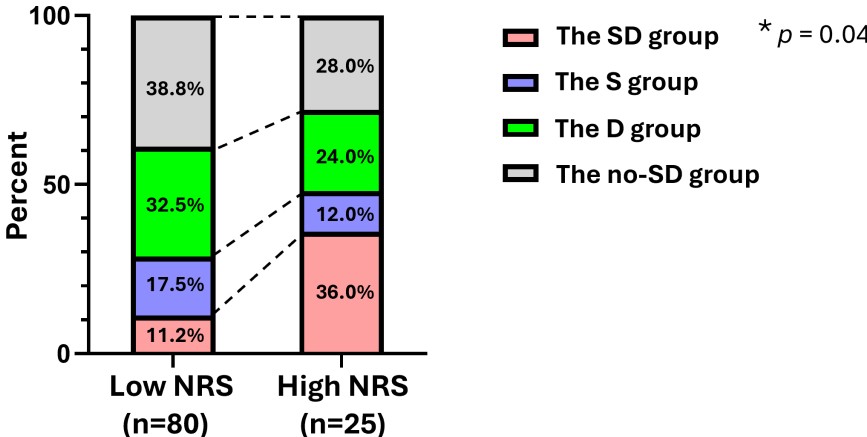

**Fig 4. The percents of the SD, S, D, and No-SD groups in the high and low NRS groups.** The percents of the SD group, the S group, the D group and the No SD group are shown in the high NRS group (NRS: 7-10, n = 25) and the low NRS group (NRS: 0-6, n = 80). The chi-squared test was used. * $p < 0.05$.

**Table 3. Odds ratio for the severity of chronic pain relative to demographics, the self-suppressing behavioral pattern scale, and the depression scale.**

| Factor | *p* value | Odds ratio | 95%CI |
|---|---|---|---|
| Age | 0.555 | 1.021 | 0.953-1.094 |
| Sex | 0.196 | 0.389 | 0.093-1.627 |
| Facility | 0.09 | 0.363 | 0.112-1.173 |
| Self suppressing behavioral pattern and Depression | **0.004 | 8.469 | 2.011-35.664 |
| Self suppressing behavioral pattern | 0.646 | 1.457 | 0.293-7.249 |
| Depression | 0.849 | 1.128 | 0.326-3.906 |

Odds ratios for the severity of chronic pain relative to age, sex, facility, high levels on both the self-suppressing behavioral pattern scale and the depression scale, high level only on the self-suppressing behavioral pattern, and high level only on the depression scale are shown. Odds ratios were calculated using the logistic regression model (n = 105). **$p < 0.01$.

trend was observed ([Fig 5a]). However, analysis of maternal attachment scores and SAT scales showed significant correlations with the self-denial scale ($r = 0.23$, $p = 0.02$) and PTSS scale ($r = 0.3$, $p = 0.003$) ([Fig 5b]), but showed no significant correlations with other scales ([S4 Fig]). Lower maternal attachment (Score 2) was associated with higher severity in these scales. In addition, the proportions of the S, D, and SD groups were compared based on maternal attachment scores and their contribution to NRS severity. No significant associations were observed ([Fig 5c]).

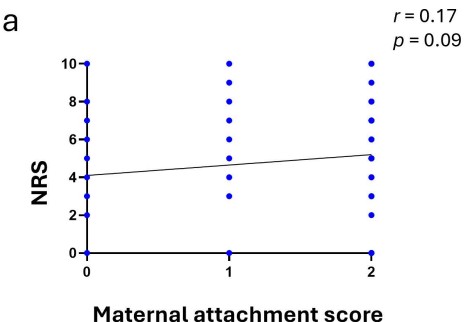

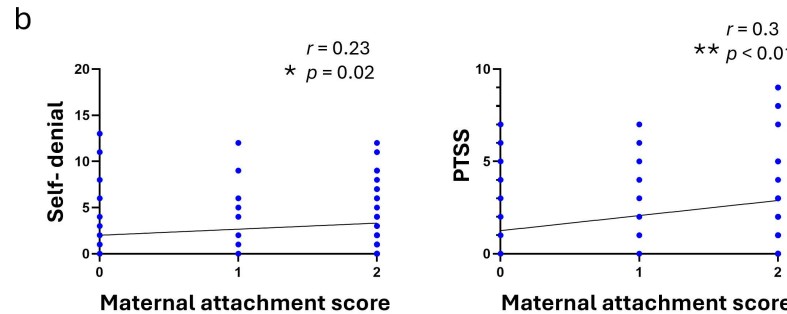

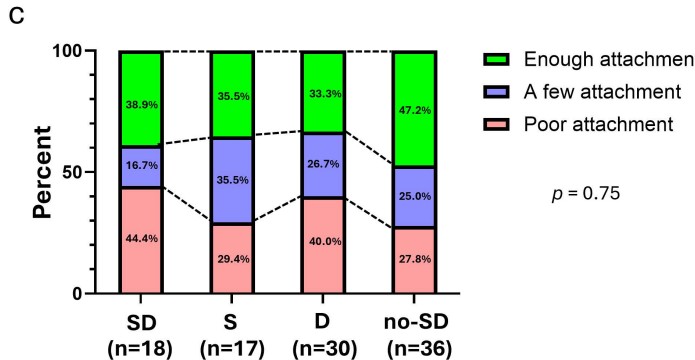

**Fig 5. The relations between the maternal attachment score and NRS or the SAT scales.** (a) The correlation between the degree of maternal attachment and the NRS score (n = 101). The scores on the maternal attachment scale are as follows: 0 = enough attachment, 1 = a little attachment, and 2 = poor attachment. Spearman's correlation coefficient was used. (b) Correlations between the maternal attachment score and the self-denial scale or the PTSS scale (n = 101). Spearman's correlation coefficient was used. *$p < 0.05$, **$p < 0.01$. The score on the maternal attachment scale was as above. (c) The percent of participants with each degree of maternal attachment for the SD group (n = 18), the S group (n = 17), the D group (n = 30), and the No-SD group (n = 36). The chi-squared test was done.

### Self-denial scale and PTSS scale in the SD group, S group, D group, and No-SD group

Last, to investigate whether the self-denial and PTSS scales were associated with the self-suppressing behavioral pattern and depression traits, these two scales were analyzed across the four groups (SD, S, D, no-SD) (Fig 6). The self-denial scale scores were significantly higher in the SD group than in the S, D, and no-SD groups. In contrast, no significant differences were observed for the PTSS scale scores across the groups.

## Discussion

This is the first study to demonstrate positive correlations between the severity of chronic pain and the self-suppressing behavioral pattern scale and the depression scale, as assessed using the SAT method. The logistic regression analysis showed that high scores on both the self-suppressing behavioral pattern scale and the depression scale were associated with a significant risk of worse chronic pain. In addition, it was found that the self-denial trait exacerbates both the self-suppressing behavioral pattern and depressive traits.

In this study, 78.1% of the participants were aged 70 years or older, and 85.7% had chronic pain. According to the Hisayama study, which is a population-based prospective study in Japan, the prevalence of chronic pain is 46.6% in individuals in their 70s and 50.9% in those in their 80s [1]. Therefore, the proportion of participants with chronic pain in this study was higher than in the general Japanese population, likely because this study targeted individuals requiring rehabilitation. In addition, 44.8% of participants in the present study had lower back pain, 33.3% had knee pain, and 21.9% had shoulder pain. Similar trends have been reported in previous studies, indicating that the primary causes of pain in individuals in their 70s and 80s in Japan are lower back pain, followed by knee pain and shoulder pain [15]. Therefore, the participants in the present study represent an appropriate cohort for analyzing the association between pain and psychological traits.

The NRS score, an objective measure of chronic pain, was correlated with both the self-suppressing behavioral pattern scale and the depression scale as assessed by the

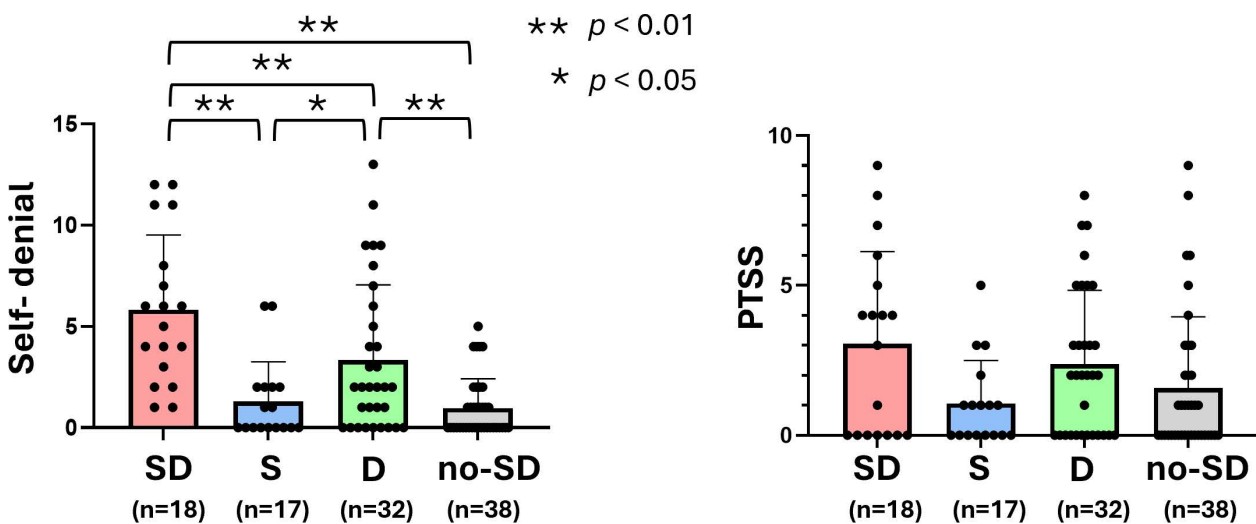

**Fig 6. Comparison of the self-denial scale and PTSS scale among the four groups (the SD group, the S group, the D group, and the No-SD group).** The comparisons of the self-denial scale, as well as the PTSS scale, among the four groups are shown (n = 105). Values are the means ± SD. The data were analyzed by one-way ANOVA followed by Tukey's post hoc comparison test. *$p < 0.05$, **$p < 0.01$.

SAT method. Participants in the high NRS group scored significantly higher on the self-suppressing behavioral pattern scale and depression scale than those in the low NRS group. Depression is commonly evaluated using established scales such as the QIDS (Quick Inventory of Depressive Symptomatology) [16] and BDI-II (Beck Depression Inventory) [17]. However, there is no global scale for assessing self-suppressing behavioral patterns. By using the SAT method, this study successfully quantified both depressive traits and self-suppressive behavioral traits, allowing for the evaluation of risk factors that exacerbate pain.

The associations between each SAT scale were previously reported. In a survey that targeted 446 university students, the depression scale was significantly correlated with the self-worth scale and the anxiety tendency scale. In addition, the anxiety tendency scale was significantly correlated with the self-worth scale, self-suppressing behavioral pattern scale, interpersonal dependency scale, and depression scale [18]. Therefore, we compared each SAT scale, focusing on the self-suppressing behavioral pattern scale and the depression scale. Both the self-suppressing behavioral pattern scale and the depression scale showed significant correlations with the self-denial scale as well as the anxiety tendency scale. However, no correlation was observed between the self-suppressing behavioral pattern scale and the depression scale. Therefore, these two scales were considered independently correlated with pain. Logistic regression analysis further confirmed that high scores on both scales significantly increase the risk of severe pain.

The self-suppressing behavioral pattern scale includes questions such as "Do you suppress your emotions?", and "Do you pay attention to what others are saying by observing their facial expressions?" [10]. This scale evaluates the difficulty of emotional expression, a characteristic commonly associated with Japanese individuals who prioritize harmony and avoid causing discomfort to others [19]. When this psychological trait becomes pronounced, it can threaten well-being and lead to mental disorders [19]. Suppressed emotions are known to be associated with increased feelings of helplessness and catastrophizing about pain [20,21]. These findings align with prior research, supporting the observed correlation between the self-suppressing behavioral pattern scale and the NRS score in the present study.

Depression or depressive state are well known to be related to chronic pain. People with severe chronic pain tend to become depressive and have a reduced quality of life [1]. In addition, chronic pain and depression are known to interact with each other, creating a vicious cycle that exacerbates both symptoms [22]. The positive correlation between the SAT depression scale and chronic pain severity observed in the present study is consistent with prior research. The finding that high scores on both the self-suppressing behavioral pattern scale and the depression scale increase the risk of more severe pain highlights the importance of assessing self-suppressive behavioral patterns in addition to depressive traits, particularly in Japanese people.

In addition, the self-denial scale was found to be associated with the degree of maternal attachment experiences in childhood. Participants with high levels on both the self-suppressing behavioral pattern scale and the depression scale also scored high on the self-denial scale. Munakata proposed that lack of attachment experiences with caregivers in childhood induces chronic stress, which can lead to chronic disease in adulthood [10]. Parental attachment is known to have a significant impact on the mental health of young adults [23]. Though this study did not find a direct correlation between the self-denial scale and the NRS or between maternal attachment experiences and the NRS, insufficient maternal attachment experiences in childhood were associated with self-denial traits. These findings suggest that poor maternal attachment may enhance self-denial traits, which in turn influence self-suppressive behavioral patterns and depressive traits, indirectly exacerbating pain (Fig 7).

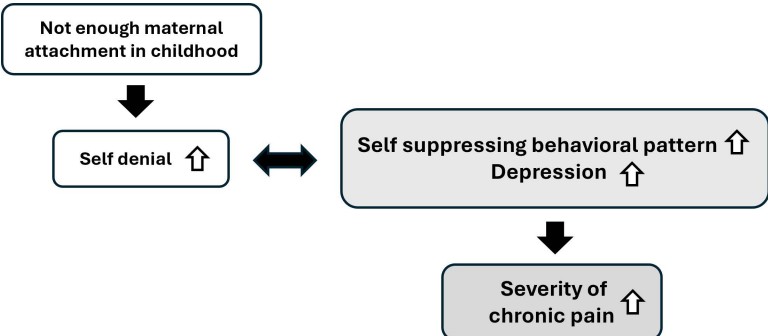

**Fig 7. The schema of psychological factors that worsen chronic pain.** Low levels of maternal attachment increase the self-denial scale. The self-denial scale is associated with both the self-suppressing behavioral pattern scale and the depression scale. Possessing both a self-suppressing behavioral pattern and depression is a potential risk factor for worse pain.

The limitations of this study include the insufficient validation of the SAT scales' reliability and validity across different ethnic groups. However, the SAT method integrates a cognitive-behavioral therapy program that addresses these psychological factors. Expanding the application of SAT therapy may contribute to improvements and the prevention of severe cases in complex conditions influenced by physical, psychological, and social factors, including chronic pain.

## Conclusion

We found that the severity of chronic pain was associated with the self-suppressing behavioral pattern scale and the depression scale. In addition, high scores on both the self-suppressing behavioral pattern scale and the depression scale were a risk of worse chronic pain. Though depression or depressive state is well known to be related to chronic pain, self-suppressing behavioral pattern was found to be an important factor, particularly in Japanese. Assessing and targeting these two factors may prevent chronic pain worsening.

## Supporting information

**S1 Table. Classification of the seven scales in SAT therapy.** The score ranges in the low, moderate, and high levels of each scale (the self-suppressing behavioral pattern scale, the depression scale, the anxiety tendency scale, the self-denial scale, the PTSS scale, the interpersonal dependency scale, and the self-worth scale) are shown.
(TIF)

**S2 Table. Demographics of the participants at each facility.** The data for age, sex, BMI, exercise time per week, NRS, duration of pain, smoking, and alcohol in the two facilities are shown. Values are the means ± SD. The unpaired *t*-test was done to compare the two facilities. **$p < 0.01$. The distribution of male and female in the two facilities was analyzed using a chi-squared test. The distribution of pain duration in the two facilities was analyzed using a chi-squared test.
(TIF)

**S3 Table. Quantification of pain areas.** The number of pain areas for each participant is counted. The number of participants with each number of pain areas is shown (n = 105).
(TIF)

**S1 Fig. The correlation between the NRS score and each SAT scale in the preliminary study.** The correlations between the NRS score and each scale in SAT therapy (the emotional support network scale (familial or not familial), the problem-solving behavioral pattern scale, the SAT therapy requirement scale, the emotional cognitive difficulty scale, the self-pity scale, and the self-dissociation scale) are shown (n = 52). Spearman's correlation coefficient was used.
(TIF)

**S2 Fig. Correlation between the NRS score and each genetic character's scale in SAT therapy.** Correlations between the NRS score and each genetic character's scale in SAT therapy (the circulatory temperament, the stickiness temperament, the autistic temperament, the obsessive temperament, the anxiety temperament, the novelty-seeking temperament) are shown (n = 92). Spearman's correlation coefficient was used.
(TIF)

**S3 Fig. Correlations between the self-suppressing behavioral pattern scale and each of the other SAT scales and correlations between the depression scale and each of the other SAT scales.** (a) Correlations between the self-suppressing behavioral pattern scale and each of the other SAT scales (except for the depression scale) (n = 105). Spearman's correlation coefficient was used. $*p < 0.05$, $**p < 0.01$. (b) Correlations between the depression scale and each of the other SAT scales (except for the self-suppressing behavioral pattern scale) (n = 105). Spearman's correlation coefficient was used. $*p < 0.05$, $**p < 0.01$.
(TIF)

**S4 Fig. Correlations between the maternal attachment score and each SAT scale.** Correlations between the maternal attachment score and the SAT scales (the self-suppressing behavioral pattern scale, the anxiety tendency scale, the depression scale, and the interpersonal dependency scale) are shown (n = 101). Spearman's correlation coefficient was used. Each score on the maternal attachment scale is as follows: 0 = enough attachment, 1 = a little attachment, 2 = poor attachment.
(TIF)

## Acknowledgments

We express our gratitude to all participants for their invaluable contributions to this study.

## Author contributions

**Conceptualization:** Shin Hashizume, Masako Nakano, Mineko Fujimiya.

**Data curation:** Shin Hashizume, Masako Nakano, Chihiro Ikehata.

**Formal analysis:** Shin Hashizume, Masako Nakano, Nobuaki Himuro.

**Methodology:** Shin Hashizume, Masako Nakano, Nobuaki Himuro.

**Supervision:** Kanna Nagaishi, Mineko Fujimiya.

**Visualization:** Masako Nakano.

**Writing – original draft:** Shin Hashizume.

**Writing – review & editing:** Masako Nakano, Kanna Nagaishi, Mineko Fujimiya.

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
