## [Decision Letter · Decision Letter 0]

27 Jan 2025

PONE-D-24-59067Self-Suppressing Behavioral Patterns and Depressive Traits Exacerbate Chronic Pain: Psychological Trait Assessment Using the Structured Association Technique MethodPLOS ONE

Dear Dr. Nakano,

Thank you for submitting your manuscript to PLOS ONE. After careful consideration, we feel that it has merit but does not fully meet PLOS ONE’s publication criteria as it currently stands. Therefore, we invite you to submit a revised version of the manuscript that addresses the points raised during the review process.

The two reviewers addressed several major and minor concerns about your manuscript. Please revise your manuscript according reviewer's comments.

We look forward to receiving your revised manuscript.

Kind regards,

Kenji Hashimoto, PhD

Section Editor

PLOS ONE

Reviewers' comments:

Reviewer's Responses to Questions

**Comments to the Author**

1. Is the manuscript technically sound, and do the data support the conclusions?

Reviewer #1: Yes

Reviewer #2: Partly

2. Has the statistical analysis been performed appropriately and rigorously? 

Reviewer #1: Yes

Reviewer #2: N/A

3. Have the authors made all data underlying the findings in their manuscript fully available?

Reviewer #1: Yes

Reviewer #2: Yes

4. Is the manuscript presented in an intelligible fashion and written in standard English?

Reviewer #1: Yes

Reviewer #2: Yes

5. Review Comments to the Author

Reviewer #1: Page 2, Line 8: Please provide specific values for the significant positive correlations.

Page 6, Line 9: Please specify the inclusion and exclusion criteria.

Page 8, Line 3: You mentioned "the preliminary analysis suggested no association with." Please support this with references.

Page 10: There is insufficient demographic information about the participants. Is there more detailed information available?

Page 12, Line 10: "Fig 1" seems to be in the wrong position.

Page 13, Line 12: Is there any missing content here?

The study lacks a conclusion and a conflict of interest section.

Reviewer #2: This study showed that the relationship between psychological traits and chronic pain using the Structured Association Technique (SAT) method to evaluate psychological factors associated with chronic pain.

The participants included 105 older adults (23 men, 82 women, mean age 80.82 years) who received rehabilitation services.

Chronic pain severity was assessed using a numerical rating scale (NRS), and psychological traits were evaluated by SAT.

In addition, maternal attachment experiences in childhood were examined. The NRS showed significant positive correlations with the self-suppressing behavioral pattern (S) scale and the depression (D) scale.

The proportion of participants with high scores on both the S and D scales (SD group) was notably higher in the high NRS group.

Logistic regression analysis showed that the SD group had a higher odds ratio (OR = 8.469, p = 0.004) for severe chronic pain, suggesting that SD traits independently contribute to worse pain.

In the SD group, the self-denial scale scores were high, and self-denial traits showed a negative correlation with maternal attachment experiences in childhood. This finding indicates that poor maternal attachment may enhance self-denial traits, which in turn indirectly worsen pain through their effects on S and D traits.

The authors concluded that the importance of S and D traits as psychological factors in chronic pain, particularly in Japanese populations, and suggest that assessing self-suppressing behavioral patterns may be beneficial for pain management.

This study is very interesting, However I have a few questions.

#1:Would you tell me about the specific physical diagnosis of the participants?

#2:Would you tell me the definition of attachment in this study?

6. PLOS authors have the option to publish the peer review history of their article (what does this mean?). If published, this will include your full peer review and any attached files.

Reviewer #1: No

Reviewer #2: No

---

## [Author Response · Author response to Decision Letter 0]

3 Feb 2025

Reviewer 1

Q1: Page 2, Line 8: Please provide specific values for the significant positive correlations.

A1: We added the specific values in Page 2, Line 8-10.

Q2: Page 6, Line 9: Please specify the inclusion and exclusion criteria.

A2: We specified the inclusion and exclusion criteria in Page 6, Line 13-16.

Q3: Page 8, Line 3: You mentioned "the preliminary analysis suggested no association with." Please support this with references.

A3: The preliminary analysis was shown in S1 Fig. We added this reference in Page 8, Line 3-6.

Q4: Page 10: There is insufficient demographic information about the participants. Is there more detailed information available?

A4: We added demographic information in Table1, dividing the participants into without chronic pain group and chronic pain group (Page 10, Line 10 and Page 11, and Line 1-2). We also added the data of duration pain, smoking and alcohol in Table1 and S2 Table (Page 25, Line 30 and Page 26, and Line 1-4).

Q5: Page 12, Line 10: "Fig 1" seems to be in the wrong position.

A5: We wrote "Fig 1" in the correct position in Page 13, Line 9-13.

Q6: Page 13, Line 12: Is there any missing content here?

A6: We wrote the content of Fig 3 in Page 14, Line 18-19 and Page 15, Line 1.

Q7: The study lacks a conclusion and a conflict of interest section.

A7: We wrote a conclusion in Page 23, Line 10-16, and a conflict of interest section in Page 28, Line 10-12.

Reviewer 2

Q1: Would you tell me about the specific physical diagnosis of the participants?

A1: We added the physical diagnosis in Table 2 (Page 12, Line 1-7). The distribution of the diagnosed and not diagnosed groups in the high and low NRS groups were also analyzed using a chi-squared test.

Q2: Would you tell me the definition of attachment in this study?

A2: We added a description of attachment in Page 8, Line 9-11.

---

## [Decision Letter · Decision Letter 1]

6 Feb 2025

Self-Suppressing Behavioral Patterns and Depressive Traits Exacerbate Chronic Pain: Psychological Trait Assessment Using the Structured Association Technique Method

PONE-D-24-59067R1

Dear Dr. Nakano,

We’re pleased to inform you that your manuscript has been judged scientifically suitable for publication and will be formally accepted for publication once it meets all outstanding technical requirements.

Kind regards,

Kenji Hashimoto, PhD

Section Editor

PLOS ONE

Additional Editor Comments (optional):

Reviewers' comments:

Reviewer's Responses to Questions

**Comments to the Author**

1. If the authors have adequately addressed your comments raised in a previous round of review and you feel that this manuscript is now acceptable for publication, you may indicate that here to bypass the “Comments to the Author” section, enter your conflict of interest statement in the “Confidential to Editor” section, and submit your "Accept" recommendation.

Reviewer #1: All comments have been addressed

Reviewer #2: All comments have been addressed

2. Is the manuscript technically sound, and do the data support the conclusions?

Reviewer #1: Yes

Reviewer #2: Partly

3. Has the statistical analysis been performed appropriately and rigorously? 

Reviewer #1: Yes

Reviewer #2: N/A

4. Have the authors made all data underlying the findings in their manuscript fully available?

Reviewer #1: Yes

Reviewer #2: Yes

5. Is the manuscript presented in an intelligible fashion and written in standard English?

Reviewer #1: Yes

Reviewer #2: Yes

6. Review Comments to the Author

Reviewer #1: The authors have addressed all suggestions made in the first round of review. In view of this, the manuscript is acceptable. -

Reviewer #2: Thank you for your polite reply.

Q1: Would you tell me about the specific physical diagnosis of the participants?

A1: We added the physical diagnosis in Table 2 (Page 12, Line 1-7). The distribution of the diagnosed and not diagnosed groups in the high and low NRS groups were also analyzed using a chi-squared test.

Q2: Would you tell me the definition of attachment in this study?

A2: We added a description of attachment in Page 8, Line 9-11.

7. PLOS authors have the option to publish the peer review history of their article (what does this mean?). If published, this will include your full peer review and any attached files.

Reviewer #1: No

Reviewer #2: No

---

## [Editor Report · Acceptance letter]

PONE-D-24-59067R1

PLOS ONE

Dear Dr. Nakano,

I'm pleased to inform you that your manuscript has been deemed suitable for publication in PLOS ONE. Congratulations! Your manuscript is now being handed over to our production team.

Kind regards,

on behalf of

Prof. Kenji Hashimoto

Section Editor

PLOS ONE